# Exploration of Medical Trajectories of Stroke Patients Based on Group-Based Trajectory Modeling

**DOI:** 10.3390/ijerph16183472

**Published:** 2019-09-18

**Authors:** Ting-Ying Chien, Mei-Lien Lee, Wan-Ling Wu, Hsien-Wei Ting

**Affiliations:** 1Department of Computer Science and Engineering, Yuan Ze University, Taoyuan City 320, Taiwan; 2Graduate Program in Biomedical Informatics, Yuan Ze University, Taoyuan City 320, Taiwan; 3Innovation Center for Big Data and Digital Convergence, Yuan Ze University, Taoyuan City 320, Taiwan; 4Department of Neurosurgery, Taipei Hospital, Ministry of Health and Welfare, New Taipei City 242, Taiwan

**Keywords:** group-based trajectory modeling, ischemic stroke, medical expenditure, national health insurance research database, spontaneous intracerebral hemorrhage

## Abstract

A high mortality rate is an issue with acute cerebrovascular disease (ACVD), as it often leads to a high medical expenditure, and in particular to high costs of treatment for emergency medical conditions and critical care. In this study, we used group-based trajectory modeling (GBTM) to study the characteristics of various groups of patients hospitalized with ACVD. In this research, the patient data were derived from the 1 million sampled cases in the National Health Insurance Research Database (NHIRD) in Taiwan. Cases who had been admitted to hospitals fewer than four times or more than eight times were excluded. Characteristics of the ACVD patients were collected, including age, mortality rate, medical expenditure, and length of hospital stay for each admission. We then performed GBTM to examine hospitalization patterns in patients who had been hospitalized more than four times and fewer than or equal to eight times. The patients were divided into three groups according to medical expenditure: high, medium, and low groups, split at the 33rd and 66th percentiles. After exclusion of unqualified patients, a total of 27,264 cases (male/female = 15,972/11,392) were included. Analysis of the characteristics of the ACVD patients showed that there were significant differences between the two gender groups in terms of age, mortality rate, medical expenditure, and total length of hospital stay. In addition, the data were compared between two admissions, which included interval, outpatient department (OPD) visit after discharge, OPD visit after hospital discharge, and OPD cost. Finally, the differences in medical expenditure between genders and between patients with different types of stroke—ischemic stroke, spontaneous intracerebral hemorrhage (sICH), and subarachnoid hemorrhage (SAH)—were examined using GBTM. Overall, this study employed GBTM to examine the trends in medical expenditure for different groups of stroke patients at different admissions, and some important results were obtained. Our results demonstrated that the time interval between subsequent hospitalizations decreased in the ACVD patients, and there were significant differences between genders and between patients with different types of stroke. It is often difficult to decide when the time has been reached at which further treatment will not improve the condition of ACVD patients, and the findings of our study may be used as a reference for assessing outcomes and quality of care for stroke patients. Because of the characteristics of NHIRD, this study had some limitations; for example, the number of cases for some diseases was not sufficient for effective statistical analysis.

## 1. Introduction

Acute cerebrovascular disease (ACVD), or stroke, incurs substantial medical expenditure [1,2]. In general, stroke can be classified into four types: ischemic stroke, spontaneous intracerebral hemorrhage (sICH), subarachnoid hemorrhage (SAH), and transient ischemic attack (TIA). Among the leading causes of hospitalization, stroke is in the top five causes of cerebrovascular hospitalization, and 10–35% of ACVD patients are of the sICH type [3,4,5,6]. The incidence differs in men and women, and female patients are generally older than male patients [4]. Stroke incurs a high medical expenditure, in particular attracting high costs for emergency and intensive care facilities in cases of sICH [2,7]. Many factors affect the outcome of ACVD, and comorbidities are critical risk factors. Ensuring continuity of care for stroke patients with comorbidities can significantly reduce their risks and improve their treatment outcomes [2,8,9]. Several studies have attempted to use cross-sectional data to predict the outcomes of strokes [10,11,12], with limited results.

Group-based trajectory modeling (GBTM) is a method of longitudinal data analysis, and can be used to evaluate the outcomes of diseases over time [13,14]. Research into sleep behavior has been performed using GBTM, and it was found that night-waking in children is positively associated with emotional symptoms, hyperactivity/inattention, and conduct problems [15]. Another study used longitudinal transition trajectory data for gout and its comorbidities for outcome prediction in different age groups [16]. The National Health Insurance Research Database (NHIRD) samples longitudinal data of National Health Insurance patients in Taiwan for healthcare research [17]. Many researchers use the NHIRD as a research database for model construction or knowledge discovery [5,7,18,19,20,21,22]. The NHIRD is a very important medical administrative database, and has been instrumental in increasing our knowledge in the areas of medicine, health, and epidemiology. This study employed GBTM to explore the trends in hospitalization in groups of stroke patients using the NHIRD.

## 2. Materials and Methods

### 2.1. Data Source, Protection, and Permissions

Patient data were collected from the NHIRD in Taiwan [17], a database of 1 million sampled cases having been extracted from the NHIRD. Data analysis was performed using a big data analytics platform in the Innovation Center for Big Data and Digital Convergence, Yuan Ze University [23]. Patient privacy was protected using a double-scrambling protocol that encrypted patient data in the NHIRD to generate data for research purposes. The study was approved by the Institutional Review Board (IRB) of Taipei Hospital (IRB Approval Number: TH-IRB-0015-0003), and all the researchers agreed to and signed a written statement that proclaimed they had no intention of acquiring information that might potentially breach the privacy of patients or care providers. The protocol of this study was assessed by the National Health Research Institutes (NHRI), which agreed to the proposed analysis of the database (Approval No.: NHIRD-104-183).

### 2.2. Data Management and Statistical Analysis

Three kinds of stroke and related diseases were identified using the International Classification of Diseases 9th revision (ICD9) principal diagnosis codes: ischemic stroke (ICD9 codes 433 to 434, 436); sICH (ICD9 codes 431 to 432); and SAH (ICD9 code 430). There were 27,646 patients admitted due to stroke from 2000 to 2013. The patients who were admitted due to suspected traumatic brain injury (ICD9 codes 800 to 804, 850 to 854, 959.01, and 959.09) and traffic events were excluded. To avoid the effect of extreme values, cases in the top 10% in terms of total length of stay and patients hospitalized more than nine times (fewer than 15 cases, about 0.05%), totaling 40 cases, were excluded from the analysis (see Table 1). In addition, we integrated the data with the registry of beneficiaries file, and identified 6946 patients not covered by National Health Insurance, which meant that the patient was dead [24], and with the data on ambulatory care expenditures by visits. Finally, these data were analyzed in a merged database. Figure 1 shows the flowchart of data management of this study.

Patient characteristics, including age, mortality rate, medical expenditure, and total length of hospital stay, were recorded, and the information for each group and each hospitalization was then analyzed. We identified patients who had been hospitalized more than four times and fewer than or equal to eight times, and their data were subjected to trajectory analysis. This study chose only the patients who had been admitted to hospitals four times because the GBTM method analyzes the trend of admissions. If there are fewer than three admissions, no significant trend patterns will be found. Data classification was based on the parameter of medical expenditure, using which patients were split into three groups: high, medium, and low medical expenditure, split at the 33rd and 66th percentiles. Data management and statistical analysis were performed using SAS software (ver.9.4; SAS Institute Inc., Cary, NC, USA).

## 3. Results

In total, there were 27,404 cases admitted to hospitals. Cases who were admitted to hospitals fewer than four times or more than eight times were excluded. A total 27,264 cases (male/female = 15,972/11,392) were included in this study (Table 1), and 795 patients had been admitted to hospital more than four times. According to our analysis, 20,713 cases were of ischemic stroke, 5758 were sICH cases, and 875 were SAH cases. The mortality rate at the initial admission (7.0%) was relatively higher than at the second (6.1%), third (6.3%), and fourth admissions (6.2%) (*p* < 0.001). The mortality rate differed between male and female patients. The mortality rate of female patients decreased from 7.2% at the first admission to 4.5% at the fourth admission. In addition, despite a relatively high mortality rate at the first admission (6.9%), the rate in male patients increased at every admission subsequent to the second (5.0% at the second, 6.1% at the third, and 7.1% at the fourth admission). With regards to the type of stroke, the mortality rate for SAH (21.1%) was significantly higher than those for sICH (13.9%) and ischemic stroke (4.5%) (*p* < 0.001). There were no significant differences in mortality rate between the first and the fourth admission for any type of stroke. The mean age of the male and female patients differed significantly at every admission, with the exception of the patients who died at the fourth admission, there being no significant difference in mean age between the male and female patients who died at the fourth admission. The mean ages of patients with sICH (58.9 years, SD = 15.2), and SAH (54.7 years, SD = 14.7) were significantly lower than the mean ages of the ischemic stroke patients at the first admission (*p* < 0.001). The same trend was observed in the patients who died, but there were no significant differences in age between the patients who suffered different types of stroke. The mean length of hospital stay (LOS) increased significantly from the first to the fourth admission in both male and female patients. However, the LOS at the third and fourth admissions were significantly decreased in the patients who died, and this was especially evident in the female patients at the fourth admission. The total medical expenditure for the patients who died was significantly higher than the mean total expenditure, regardless of admission or gender (*p* < 0.001). sICH (16.5 days, SD = 13.5) and SAH patients (17.0 days, SD = 13,8) had significantly longer LOS than ischemic stroke patients (9.1 days, SD = 6.7) at the first admission (*p* < 0.001). In contrast, of the patients who died, the SAH (9.6 days, SD = 11.0) cases, ischemic stroke cases (10.5 days, SD = 8.2), and sICH cases (11.3 days, SD = 12.5) did not differ significantly in terms of LOS at the first admission. At the different admissions, the mean medical expenditure of the patients who died was significantly higher than that of the patients who did not expire (*p* < 0.001), and decreased from the first to the fourth admission, regardless of gender or stroke type. Only in the SAH cases was the medical expenditure of the patients who died ($4932, SD = 4471) significantly lower than that of the other patients ($7636, SD = 6373) at the first admission. However, according to the different types of stroke, the medical expenditure of the SAH patients ($7636, SD = 6373) was significantly higher than that of the sICH ($3846, SD = 3776) and ischemic stroke patients ($1293, SD = 1061) at the first admission. Interestingly, the medical expenditure of the patients who died was lower than that of the total patient cohort at the fourth admission (Table 2).

Using the GBTM method, the data were compared between two admissions. The interval between two admissions decreased from 522.8 days (SD = 598.3) to 364.6 days (SD = 426.1) and to 316.5 days (SD = 370.6). There was no significant difference in the interval between the first and second admissions between the male (536.7 days, SD = 597.5) and female patients (543.5 days, SD = 599.6). However, the interval between the second and third admissions was longer in the female patients (428.6 days, SD = 484.7) than the male patients (356.7 days, SD = 393.4) (*p* < 0.001). The interval between the first and second admissions in the ischemic stroke patients (556.2 days, SD = 602.1) was significantly longer than that of the sICH (473.9 days, SD = 564.4) and SAH patients (474.5 days, SD = 614.4) (*p* < 0.001). Interestingly, the interval between the third and fourth admissions in the SAH cases (188.6 days, SD = 119.3) was significantly shorter than that in the ischemic stroke (335.1 days, SD = 370.4) and sICH patients (333.8 days, SD = 378.2) (*p* < 0.001). There was no significant difference in the interval between the third and fourth admissions between the male (336.1 days, SD = 317.5) and female patients (328.2 days, SD = 317.5). The first outpatient department (OPD) visit after discharge occurred around 13 days after the first admission in the male patients, and 11 days in the female patients. In the different admission intervals, the highest mean number of OPD visits occurred between the second and third admissions (68.9 times, SD = 70.4) as compared with the first and second (45.1 times, SD = 50.5) and the third and fourth admissions (56.7 times, SD = 60.3), regardless of gender. Patients visited the OPD around every 5 days, incurring a cost of approximately $31 to $37 per visit, regardless of gender or type of stroke (Table 3).

This study evaluated the differences between genders in medical expenditure at different hospital admissions using GBTM. The cases were divided into high, medium, and low groups according to medical expenditure, separated at the 33rd and the 66th percentiles; a total 28.8% of cases incurred a high medical expenditure and 37.9% cases a low medical expenditure at the first admission; however, from the second to the fourth admission, the high medical expenditure group increased in size progressively from 32.7% to 33.9% and finally to 35.7% in the male patients, and 34.3% to 34.0% and finally to 37.5% in the female patients. The percentage of patients in the medium medical expenditure group remained stable in both genders. More than 26% of patients migrated from the low medical expenditure group to the high group in both the female and male patients at subsequent admissions (Figure 2).

This study also evaluated the differences in medical expenditure between patients with different types of stroke at different hospital admissions using GBTM. According to the different types of stroke, the medical expenditure at different admissions differed. The ischemic stroke patients (644 cases) represented more than 80% of all stroke cases analyzed, while the sICH patients accounted for 141 cases and the SAH patients for only 10 cases of all the stroke patients who had been admitted to hospital more than four times. As the number of cases was relatively low, GBTM was used to evaluate the SAH patients for reference only. The proportions of sICH (46.1%) and SAH (90%) patients with a high medical expenditure were relatively higher than the proportion of ischemic stroke patients (23.2%), and the proportion of ischemic stroke cases was relatively higher than the proportions of sICH (29.0%) and SAH (0.0%) patients in the low medical expenditure group at the first admission. More than 45% of patients in the high medical expenditure group and more than 34% in the medium medical expenditure group had suffered the same type of stroke. The proportion of remaining ischemic stroke patients in the low medical expenditure group decreased progressively from 50.4% to 42.0% and finally to 38.0%, but the proportion of patients who moved into the higher medical expenditure group increased at subsequent admissions, regardless of stroke type (Figure 3).

## 4. Discussion

Group-based trajectory modeling (GBTM) is a good method by which to examine trends in different groups [13,14,25], sometimes uncovering as yet undiscovered knowledge. This study used GBTM to examine the trends in medical expenditure in different groups of ACVD patients at different admission times, and some important results were obtained. Although at the first admission, the SAH (90%) and sICH patients (46.1%) incurred higher medical expenditures than the ischemic stroke patients (23.2%), the results of this study showed that many ACVD patients in the low or medium expenditure groups moved to the high expenditure group at subsequent admissions, indicating that medical expenditure will increase at each subsequent hospitalization for ACVD in the majority of patients. Our study results also demonstrated that the time interval between two admissions became progressively shorter; this phenomenon was observed in both genders, as well as for all types of stroke.

Even though it is generally agreed that the state of patients will worsen gradually with age, questions remain as to whether or not patients with ACVD will become ill again, and whether hospitalization will occur within a shorter interval than for previous admissions. More studies are needed in order to investigate these critical questions. At present, an inadequate quality of care that results in worsened patient condition and an elevated medical expenditure in ACVD patients may be a possible explanation for this observation.

There are differences between male and female patients in many diseases. Gender is a special factor in terms of the outcome of ACVD. Women have a lower incidence of ischemic stroke as compared with men throughout most of their lifespan [26]. In fact, gonadal hormones are an important factor in coagulation and fibrinolysis, and influence the risk of ischemic stroke [26,27]. However, women are at greater risk of sICH than men [4]. Women will more frequently ignore symptoms of acute ischemic stroke, leading to a delay in arriving at a hospital [28]. This study found that there was no significant difference in mortality rate between male and female stroke patients at the first admission. Interestingly, the trend of female mortality was that of a progressive decrease, especially at the fourth admission. In contrast, the trend of male mortality was an increase from the second admission. This might be due to the fact that in female stroke patients, the first attack occurs at an older age, while male stroke patients are relatively younger than female patients. This was very interesting, and further study is needed to investigate the reasons for these results.

It is difficult to make the decision as to when to stop treatment and allow patients to die with dignity. The old patients with preoperative do-not-resuscitation (DNR) orders is significant associated with postoperative morbidity and mortality [29]. Stroke patients are at high risk of morbidity and mortality. One study indicated that 35% of sICH patients would consider a DNR order, and 73% of orders were issued within 24 h of admission. A more severe stroke, greater age, and deterioration soon after admission were three of the most important factors [30]. The medical expenditure of patients who died was significantly higher than the total mean medical expenditure at the first admission, and decreased at subsequent hospital admissions. Cases of SAH and sICH, which are more severe types of stroke than ischemic stroke, incurred greater medical expenditures, which decreased significantly at the second, third, and fourth admissions. As in previous studies of stroke patients, these results demonstrated that different decisions are made if patients have multimorbidity, such as a more severe stroke, an older age, another stroke, or deterioration soon after admission [30]. DNR orders are positively associated with multimorbidity, cognitive impairment, cancer, and stroke [31]. Patients themselves or their families will consider conservative treatment and a DNR order when patients become more poorly and are admitted again. This represents indirect evidence that patients who do not recover from diseases will tend to sign a DNR order to ensure a better quality of life and to prevent invalidation of medical treatment in Taiwan. Patients who suffer a stroke need better care and a better quality of life, which would be expected to result in postponing the next admission and allow the patient dignity in death. The results of this study may be used as a reference in terms of evaluation of outcomes and quality of care for stroke patients.

This study had some limitations. First, the data used in this study were administrative data, which may represent patient outcome indirectly. Second, this study chose patients who had been admitted to hospitals more than four times, as the GBTM method used evaluates the trend of admissions. If there are less than three admissions, no significant trend patterns will be found. This method also had bias for the cases with more than 8 admissions because of that there were not enough cases to derive a trend for them.Third, the patient data were derived from 1 million sampled cases from the NHIRD; therefore, the number of cases for some diseases was not sufficient for effective statistical analysis; for example, there were only 10 SAH cases analyzed in this study. Some trends or mean data were relatively extreme as compared with other groups, and the results for SAH have been included only for reference. Further evaluation is needed. Fourth, some patients are hospitalized for many diseases, and, in order to simplify the statistics, comorbidities were not evaluated in this study, which may have caused some bias. In the future, we might use multi-hospital data and big data methodologies for evaluation. We might also use the whole NHIRD, and comorbidities and outcome factors will be considered in future studies.

## 5. Conclusions

This study employed GBTM to examine the trends in medical expenditure for different groups of stroke patients at different admissions, and some important results were obtained. Our results demonstrated that the time interval between subsequent hospitalizations decreased in the ACVD patients, and there were significant differences between genders and between patients with different types of stroke. It is often difficult to decide when the time has been reached at which further treatment will not improve the condition of ACVD patients, and the findings of our study may be used as a reference for assessing outcomes and quality of care for stroke patients. Because of the characteristics of NHIRD, this study had some limitations; for example, the number of cases for some diseases was not sufficient for effective statistical analysis.

## Figures and Tables

**Figure 1 ijerph-16-03472-f001:**
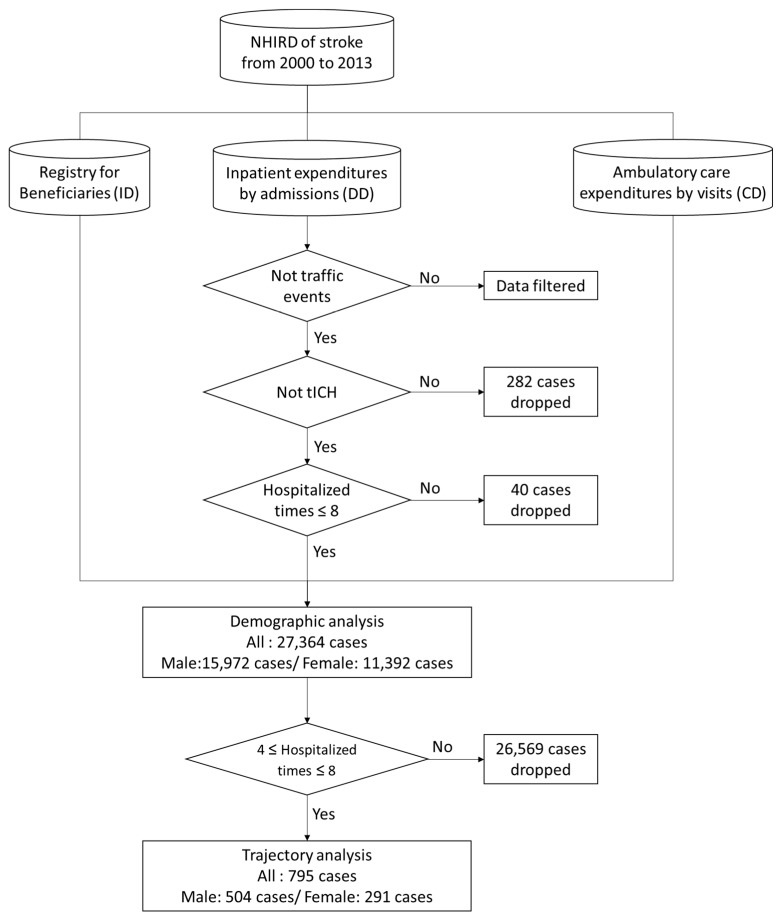
Flow chart describing management of data from the National Health Insurance Research Database in Taiwan. tICH: Traumatic Intracranial Hemorrhage.

**Figure 2 ijerph-16-03472-f002:**
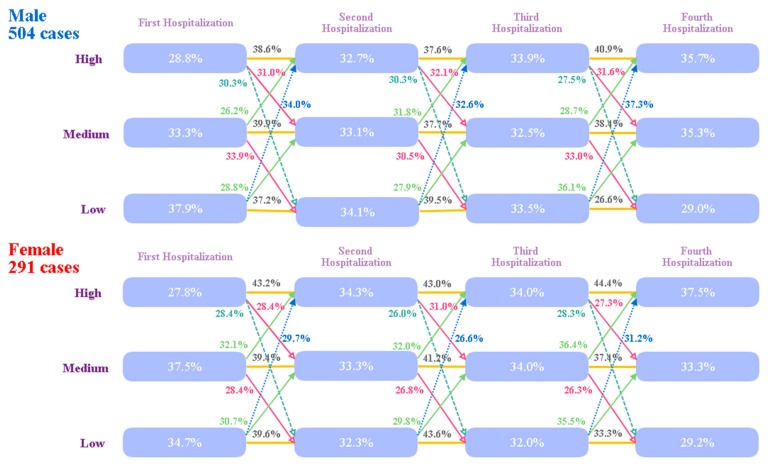
Group-based trajectory model (GBTM) of medical expenditure of stroke patients of different genders. Patients were divided into three groups according to medical expenditure, high, medium, and low groups, separated at the 33rd and the 66th percentiles. The trend of movement of patients to groups of higher medical expenditure was observed in both genders.

**Figure 3 ijerph-16-03472-f003:**
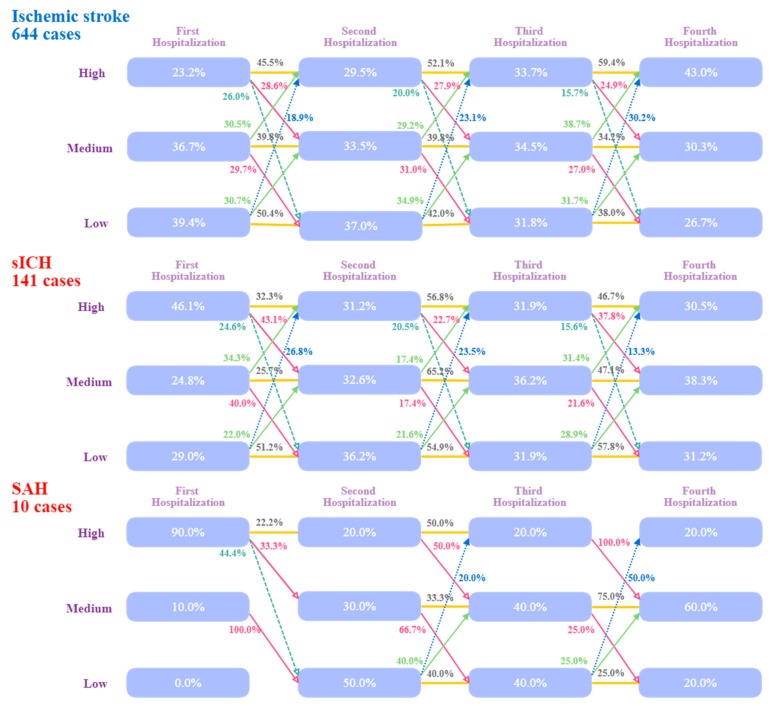
Group-based trajectory model (GBTM) of medical expenditure of stroke patients according to type of stroke. As the number of cases was relatively low, GBTM was used to evaluate SAH cases for reference only. The proportions of sICH and SAH patients in the high medical expenditure group were relatively higher than the proportion of ischemic stroke patients at the first admission. Patients increasingly moved to groups of higher medical expenditure at subsequent admissions, regardless of stroke type.

**Table 1 ijerph-16-03472-t001:** Patient hospitalization statistics.

Admission Times (*n*)	# Patients	Percentage	# Patients (*n* ≤ Admission Times ≤ 8)
1	20,392	74.41%	27,364
2	4772	17.41%	6972
3	1405	5.13%	2200
4	497	1.81%	795
5	183	0.67%	298
6	68	0.25%	115
7	28	0.10%	47
8	19	0.07%	19
9	14	0.05%	
10	7	0.03%	
>10	19	0.07%	
Total	27,404		

**Table 2 ijerph-16-03472-t002:** Demographic data at different hospital admissions.

	First Hospitalization		Second Hospitalization		Third Hospitalization		Fourth Hospitalization	
	All	Death	All	Death	All	Death	All	Death
§ Case numbers (%)	27,364	1924 (7.0%)	6972	428 (6.1%)	2200	138 (6.3%)	795	49 (6.2%)
Male	15,972	1100 (6.9%)	4130	224 (5.0%)	1351	82 (6.1%)	504	36 (7.1%)
Female	11,392	824 (7.2%)	2842	204 (7.2%)	849	56 (6.6%)	291	13 (4.5%)
Ischemic stroke	20,731	937 (4.5%)	5579	345 (6.2%)	1782	101 (5.7%)	644	40 (6.2%)
sICH	5758	802 (13.9%)	1263	76 (6.0%)	385	33 (8.6%)	141	9 (6.4%)
SAH	875	185 (21.1%)	130	7 (5.4%)	33	4 (12.1%)	10	0 (0.0%)
₸ Mean age (years)	64.9 ***	64.6 ***	67.3 ***	69.7 *	68.1 ***	68.8 ***	67.7 ***	70.8
(SD)	(13.3)	(15.1)	(12.2)	(12.2)	(11.8)	(11.5)	(11.6)	(11.5)
Male	63.1	62.1	66.0	67.8	66.5	66.4	66.9	71.7
(SD)	(13.1)	(14.6)	(12.2)	(12.0)	(11.6)	(11.5)	(11.5)	(10.2)
Female	67.1	67.1	69.6	71.8	69.8	73.4	69.9	71.1
(SD)	(13.1)	(15.1)	(11.9)	(12.2)	(11.7)	(10.9)	(11.9)	(12.7)
Ischemic stroke	66.9	70.6	68.9	72.2	69.6	68.6	69.3	73.5
(SD)	(11.8)	(12.0)	(10.9)	(9.6)	(10.4)	(11.5)	(10.3)	(7.9)
sICH	58.9	59.0	61.2	60.2	61.3	67.3	61.3	56.7
(SD)	(15.2)	(15.4)	(14.5)	(15.9)	(14.6)	(10.6)	(14.5)	(17.2)
SAH	54.7	57.2	56.9	56.3	59.0	77.8	56.6	X
(SD)	(14.7)	(13.1)	(15.0)	(8.7)	(13.6)	(18.4)	(8.7)	X
Length of hospital stay (days)(SD)	10.6 (8.5)	10.7 (10.0)	11.0 (8.6)	11.5 * (10.1)	11.6 (8.7)	11.4 (10.3)	12.2 (9.3)	10.1 (9.0)
Male	10.4	10.6	11.0	11.3	11.7	11.8	12.5	11.1
(SD)	(8.5)	(10.2)	(8.8)	(10.5)	(8.8)	(10.6)	(9.9)	(11.5)
Female	10.9	11.0	11.1	11.6	11.6	11.0	11.7	7.8
(SD)	(8.5)	(9.6)	(8.5)	(9.4)	(8.7)	(10.0)	(8.6)	(6.6)
Ischemic stroke	9.1	10.5	10.4	11.0	11.1	11.3	11.7	9.5
(SD)	(6.7)	(8.2)	(7.9)	(9.2)	(8.3)	(10.4)	(8.7)	(7.1)
sICH	16.5	11.3	14.3	10.5	14.2	14.1	16.0	17.6
(SD)	(13.5)	(12.5)	(11.7)	(10.4)	(10.4)	(12.2)	(14.3)	(18.9)
SAH	17.0	9.6	13.0	14.0	13.7	₸ 117.5	13.7	X
(SD)	(13.8)	(11.0)	(11.1)	(12.8)	(13.3)	(201.6)	(8.8)	X
§ Medical expenditure	1806	2945	1677	2594	1662	2117	1703	1961
(SD)	(1838)	(2295)	(1556)	(1932)	(1472)	(1691)	(1504)	(1804)
Male	1751	2887	1656	2392	1645	2046	1745	2167
(SD)	(1784)	(2247)	(1565)	(1884)	(1485)	(1711)	(1623)	(1879)
Female	1883	3058	1708	2805	1688	2203	1647	1500
(SD)	(1908)	(2395)	(1544)	(1965)	(1455)	(1683)	(1345)	(1832)
Ischemic stroke	1293	2166	1534	2464	1588	2236	1665	2047
(SD)	(1061)	(1389)	(1379)	(1780)	(1384)	(1784)	(1498)	(1871)
sICH	3846	4164	2265	3188	1972	2031	1929	1424
(SD)	(3776)	(3886)	(2273)	(2783)	(1792)	(1825)	(1717)	(1338)
SAH	7636	4932	3117	8423	2180	1896	1626	X
(SD)	(6373)	(4471)	(3064)	(9799)	(2506)	(2000)	(853)	X

* *p* < 0.05, ** *p* < 0.01, *** *p* < 0.001. § Percentage of patients who died. ₸ Extreme data due to too few data being used in the calculation. sICH: spontaneous intracerebral hemorrhage; SAH: Subarachnoid hemorrhage; SD: standard deviation.

**Table 3 ijerph-16-03472-t003:** Comparison of demographic data between two hospital admissions.

§	Interval 1 to 2	§ Interval 2 to 3	§ Interval 3 to 4
Mean days between two admissions	522.8 (598.3)	364.6 (426.1)	316.5 (370.6)
Male (SD)	536.7 (597.5)	356.7 (393.4)	336.1 (317.5)
Female (SD)	543.5 (599.6)	428.6 (484.7)	328.2 (371.5)
Ischemic stroke (SD)	556.2 (602.1)	385.4 (423.3)	335.1 (370.4)
sICH (SD)	473.9 (564.4)	363.4 (438.7)	333.8 (378.2)
SAH (SD)	474.5 (614.4)	423.3 (398.7)	188.6 (119.3)
Mean OPD visits (SD)	45.1 (50.5)	68.9 (70.4)	56.7 (60.3)
Male (SD)	42.3 (47.1)	63.3 (65.6)	54.6 (58.9)
Female (SD)	49.6 (55.5)	78.0 (77.8)	60.4 (63.3)
Ischemic stroke (SD)	46.5 (50.9)	69.1 (70.7)	56.5 (59.8)
sICH (SD)	39.7 (49.1)	69.4 (71.2)	57.7 (64.2)
SAH (SD)	39.0 (48.1)	56.9 (55.1)	82.4 (90.0)
Mean days to OPD visit after hospital discharge (SD)	4.8 (3.1)	5.4 (3.8)	5.7 (4.6)
Male (SD)	4.8 (3.0)	5.4 (3.9)	5.6 (4.4)
Female (SD)	4.8 (3.1)	5.5 (3.9)	5.7 (4.7)
Ischemic stroke (SD)	4.7 (2.9)	5.2 (3.5)	5.4 (4.2)
sICH (SD)	5.5 (4.8)	6.0 (5.0)	6.1 (5.2)
SAH (SD)	7.4 (7.0)	7.7 (7.1)	10.2 (8.5)
¶ Mean OPD cost per visit (SD)	31 (26)	34 (28)	37 (30)
Male (SD)	31 (26)	35 (28)	36 (36)
Female (SD)	31 (26)	34 (27)	38 (31)
Ischemic stroke (SD)	31 (26)	34 (28)	36 (29)
sICH (SD)	32 (26)	35 (28)	40 (32)
SAH (SD)	25 (20)	31 (26)	40 (30)

§ Interval represents the data between two admissions. ¶ Medical expenditure is presented in US dollars. The ratio of US dollars to Taiwan dollars is 1:30. sICH: spontaneous intracerebral hemorrhage; SAH: Subarachnoid hemorrhage; SD: standard deviation; OPD: Outpatient department.

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
