# Peer review of "Exploration of Medical Trajectories of Stroke Patients Based on Group-Based Trajectory Modeling"

_ijerph, 2019, doi:10.3390/ijerph16183472_

Round 1

Reviewer 1 Report

This paper uses the group-based trajectory modeling (GBTM) to study medical expense incurred by hospitalization due to acute cerebrovascular disease. The patients were obtained from the National Health Insurance Research Database in Taiwan. Patients who have been hospitalized more than 3 times but less than 9 times are included and three types of stokes, including ischemic stroke, SICH, and SAH, are studied. It was found that time interval between each hospital admission decreases and there were significant differences between genders and between different stroke groups.

From the experiment design and results, I do not see how they systematically and statistically support the statement that “the findings of our study may be used as a reference for assessing the outcome and quality of care for stroke patients”. It is not clear how many patients are studied using the GBTM. From figure 1, it seems only 795 cases have at least 4 hospital admissions. There are 6946 cases with out insurance, how their expenses affect the results? Why only cases with 4 – 8 hospital admissions are included?

Reviewer 2 Report

In this report, the group-based trajectory modeling (GBTM) was used to analyze the medical expenditures associated with different forms of acute cerebrovascular disease (ACVD) from different genders, ages and number of hospitalizations. GBTM emerges as powerful longitudinal data analysis for the trends of disease progress and prognosis prediction among different groups. In this case, the study extended further to ACVD medical expenditure analysis, etc. Based on a large sample size of patients and rigorous data exclusion criteria, significant and important results are generated from this study. The limitation of the data resource is fairly acknowledged. The resultant information appears to have a valuable impact on stroke clinic treatment strategy and medical expenditure planning for ACVD.  However, the authors need to address the following issues in the revision.

Abstract: the results and conclusion parts are not informative enough. The authors should provide as much as take-home messages as possible to the readers. For example the positive association of ACVD medical expenditure with the number of subsequent admissions. The specificity of the difference in genders revealed from this analysis. Line 52: “This study used…” should read as “Another study used…” Line 76: The abbreviations for sICH (intracerebral hemorrhage), and SAH (subarachnoid hemorrhage) have to be spelled in full before use.

Author Response

This manuscript is a resubmission of an earlier submission. The following is a list of the peer review reports and author responses from that submission.